# Rapa Nui (Easter Island) Rano Raraku crater lake basin: Geochemical characterization and implications for the Ahu-Moai Period

E. Argiriadis[1,2], M. Bortolini[2]*, N. M. Kehrwald[3], M. Roman[2], C. Turetta[1], S. Hanif[2], E. O. Erhenhi[2], J. M. Ramirez Aliaga[4], D. B. McWethy[5], A. E. Myrbo[6], A. Pauchard[7], C. Barbante[1,2], D. Battistel[1,2]

1 Institute of Polar Sciences CNR-ISP, Venice, Italy, 2 Department of Environmental Sciences, Informatics and Statistics, Ca' Foscari University of Venice, Venice, Italy, 3 U.S. Geological Survey, Geosciences and Environmental Change Science Center, Denver, CO, United States of America, 4 HUB Ambiental UPLA, Universidad de Playa Ancha, Valparaíso, Chile, 5 Department of Earth Sciences, Montana State University, Bozeman, MT, United States of America, 6 Department of Earth Sciences, University of Minnesota, Twin Cities, Minneapolis, MN, United States of America, 7 Laboratorio de Invasiones Biológicas, Facultad de Ciencias Forestales, Universidad de Concepción, Concepción, Chile

* mara.bortolini@unive.it

**Data Availability Statement:** All the data collected for this research are available on Mendeley Data

## Abstract

Rano Raraku, the crater lake constrained by basaltic tuff that served as the primary quarry used to construct the *moai* statues on Rapa Nui (Easter Island), has experienced fluctuations in lake level over the past centuries. As one of the only freshwater sources on the island, understanding the present and past geochemical characteristics of the lake water is critical to understand if the lake could have been a viable freshwater source for Rapa Nui. At the time of sampling in September 2017, the maximum lake depth was ~1 m. The lake level has substantially declined in the subsequent years, with the lake drying almost completely in January 2018. The lake is currently characterized by highly anoxic conditions, with a predominance of ammonium ions on nitrates, a high concentration of organic carbon in the water-sediment interface and reducing conditions of the lake, as evidenced by Mn/Fe and Cr/V ratios. Our estimates of past salinity inferred from the chloride mass balance indicates that it was unlikely that Rano Raraku provided a viable freshwater source for early Rapa Nui people. The installation of an outlet pipe around 1950 that was active until the late 1970s, as well as grazing of horses on the lake margins appear to have significantly impacted the geochemical conditions of Rano Raraku sediments and lake water in recent decades. Such impacts are distinct from natural environmental changes and highlight the need to consider the sensitivity of the lake geochemistry to human activities.

## 1. Introduction

Hundreds of massive megalithic statues, known as *moai*, populate the landscape of Rapa Nui (Easter Island). The walls constraining the Rano Raraku crater served as the primary quarry for *moai* by the ancient Rapa Nui inhabitants between ~1200 and 1600 CE [1]. Due to the

using the following link https://data.mendeley.com/datasets/k93rp3p4pd/1.

**Funding:** This study was supported by Università Ca' Foscari di Venezia through the Progetto di Ateneo 2015 in the form of funds awarded to DB, the National Science Foundation in the form of a grant awarded to DM to cover his travel to and accommodation in Rapa Nui (BCS-1832486), CSDCO/LacCore through field activity by AM and sampling instrumentation, and Elga (High Wycombe, UK) in the form of the PURELAB Ultra system employed for producing the ultrapure water (18 MΩ cm-1 resistivity) used throughout the study and laboratory research activity at Ca' Foscari University. The funders had no role in study design, data collection and analysis, decision to publish, or preparation of the manuscript.

**Competing interests:** The authors have read the journal's policy and have the following competing interests: Elga (High Wycombe, UK) provided support in the form of the PURELAB Ultra system employed for producing the ultrapure water (18 MΩ cm-1 resistivity) used throughout the study and laboratory research activity at Ca' Foscari University. This does not alter our adherence to PLOS ONE policies on sharing data and materials. There are no other patents, products in development or marketed products associated with this research to declare.

importance of Rano Raraku to the creation of the *moai*, archeologists have intensely investigated this site [2, 3, 4–12]. These studies demonstrate a consensus on the cultural centrality of Rano Raraku from ~1200 until at least ~1450 CE. During the time when humans were carving *moai*, people cultivated taro and sweet potato in adjacent uplands [11], and palms were widespread in the landscape surrounding Rano Raraku [9]. The timing and reasons for the cessation of *moai* quarrying at Rano Raraku are hotly debated. Many explanations suggest abandonment of quarrying may have been associated with major cultural and environmental changes on the island [3, 13–18].

The Rano Raraku crater was formed ~0.7 mya along the Rano Kau ridge. The area around Rano Raraku is a single tuff cone, and in some sectors, the tuff is strongly altered to become reddish ash. The tuff is stratified, composed of sideromelane, slightly altered to palagonite, and contains lithic blocks of older lavas. Scoria lapilli are present with olivine, clinopyroxene, plagioclase, and ilmenite [19].

The crater catchment supports a lake (~80 masl) that is directly adjacent to one of the primary *moai* quarries on Rapa Nui, providing a link between archeological investigations and evidence of past environmental changes. Rano Raraku (~300 m in diameter) has a small catchment size on the scale of ~1100 m$^2$, is a closed basin with no inflow or outflow streams and is therefore an ideal archive for paleoenvironmental reconstructions [9, 13, 20–23].

Lake sediments contain multiple proxies for reconstructing past environments and geochemical conditions (e.g., charcoal, pollen, biological indicators, and elemental chemistry [24–26]. Additionally, lake level fluctuations can be reconstructed through marginal depositional facies [27]. Past variability in salinity can be inferred from Mg/Ca and Ca/Sr ratios. In more saline water, the precipitation of high-Mg aragonite (or calcite) is favored compared to low-Mg calcite, resulting in an increased Mg/Ca ratio in the sediment. As aragonite is also enriched in Sr, an increase in Mg/Ca generally coincides with a Ca/Sr decrease [26, 28–30]. Lakes respond dynamically to changes in the water balance, especially in closed systems. When evaporation increases, lake water often becomes more saline and its composition becomes enriched in the heavier oxygen and hydrogen isotopes ($\delta^{18}$O and $\delta^2$H) [31].

Lake conditions also affect the ratios of several redox sensitive elements, such as Mn/Fe [32] and Cr/V [33]. Changes in the oxidation state of Mn affect their solubility and mobility. Mn and Fe have different redox potentials, so that Mn is reduced (oxidized) faster (slower) than Fe, resulting in a decrease in Mn/Fe in the sediment when oxygen concentrations decrease in the water column [34, 35]. Chromium (Cr) and vanadium (V) exhibit a similar behavior, with distinct oxidation states as well as the tendency to form complexes with Fe-hydroxides or organic ligands, depending on their redox state [36]. As a result, Cr/V increases in sediments under anoxic conditions and decreases in oxic conditions. Rare earth elements (REEs) help determine past hydrological and transport processes inside the basin, lake level variability, and *in situ* partitioning [37]. The fractionation of light and heavy rare earth elements (LREEs and HREEs, respectively) is promoted by preferential leaching during runoff and/or the formation of compounds with different solubilities in the aquatic environment, as HREE complexes are generally more soluble than LREEs [38]. The different forms of the REE cerium can provide additional information on fractionation processes and redox conditions. Both of the oxidized forms of Ce (trivalent and tetravalent) are less soluble than the other REEs. Therefore, leaching processes can lead to fractionation because more soluble Ce species are preferentially transported into the lake where redox conditions drive the mobility cycle of cerium. In oxic conditions, Ce(IV) prevails resulting in the enrichment of the element in the sediment and subsequent depletion in the lake water [39, 40].

In this paper, we investigate the modern environmental processes at the Rano Raraku crater basin as recorded from TEs and REEs in water, soil, and surface sediments and their

application for past environmental changes. Many studies conducted on Rano Raraku sediment cores highlight the difficulties in the determination of signals from the past few centuries, after the first human settlement, as radiocarbon dating and age-depth chronologies are poorly resolved for the last 2000 years [7, 8]. Saez et al. [6] reconstructed the past hydrological changes in Rapa Nui for the last 34,000 years BP through major element quantification, but the conditions of the lake since the human occupation of the island remain unclear [6]. Macro and microfossils in soil surveys demonstrate the higher lake levels (~10 m above the current level) before the settlement period [11]. Information about the pH and electrical conductivity from previous research [41] is limited. To enrich interpretations of recent past environmental change from paleoecological records from Rano Raraku, this study provides a detailed characterization of modern lake and geochemical conditions. Moreover, the reconstruction of the lake conditions can help evaluate the likelihood that the Rano Raraku basin provided a viable freshwater resource on the island. Our detailed characterization can also contribute to a better understanding of the interconnection between environment and socio-cultural transformation from the Ancient Moai (beginning c. 1000 to 1100 CE) to the Birdman cult (c. 1600 CE) cultural periods.

## 2. Materials and methods

### 2.1 Study site

Rapa Nui (Easter Island) is a small island that rises above sea level (~160 km$^2$) from a large volcanic complex located in the Pacific Ocean (27˚09'S, 109˚26'W) (Fig 1). Rapa Nui is one of the most isolated inhabited islands in the world as the nearest mainland is the Chilean coast, more than 3,500 km away. The climate is warm-temperate, with remarkably stable seasonal and daily temperatures ranging from mean ~18˚C in July-September to mean temperatures of ~23˚C in January-March. Mean annual precipitation is ~1,070 mm, and monthly precipitation amounts are slightly more abundant in April-June (mean monthly values >100 mm) than other months of the year. The rainfall distribution significantly differs across the island, with higher precipitation at higher elevations (e.g., Mount Terevaka, 507 masl) and lower precipitation in lowlands including Hanga Roa (12 masl), the primary town on Rapa Nui [42]. Rainfall quickly infiltrates volcanic soils on Rapa Nui and, as a result, few permanent streams or surface water bodies exist on the island. Today, Rano Raraku (Fig 1) is one of the three main freshwater sources of the island along with Rano Kau and Rano Aroi in crater basins. The crater (Figs 1 and 2B) is >300,000 years old [43] and consists of basaltic lapilli tuff used in the statues [44]. The chemistry and petrography of basaltic rocks and tuff of the Rano Raraku quarry show a general enrichment in Fe and a depletion of alkali and silica content of ~55–60% [43, 45].

Large-scale climatic variability has acted as a primary control on crater basin lake levels and sedimentation dynamics during the last 34,000 years [6, 8]. Increased sedimentation of material largely derived from the catchment indicates strong erosion and runoff around the termination of the last glacial (34,000–14,600 cal. yr BP, [8]). Over the last 34,000 years, lake levels at Rano Raraku have fluctuated dynamically with several periods where the absence of sedimentation suggests the lake dried. Following high lake levels c. 17,000 cal yr. BP when lake levels were as high as 13 m, lake levels, while highly variable, generally declined. Warm conditions, reduced runoff, and increased productivity led to lower lake levels during the early Holocene. Lake levels decreased significantly during the middle Holocene (8700–4500 cal. yr BP), promoting the onset of anoxic conditions. Dry conditions during the late Holocene led to hiatuses in sedimentation between 4500 and 800 cal. yr BP. Sedimentation resumed after 800 cal. yr BP. However, drying may have occurred many times in the recent decades to centuries, as a result

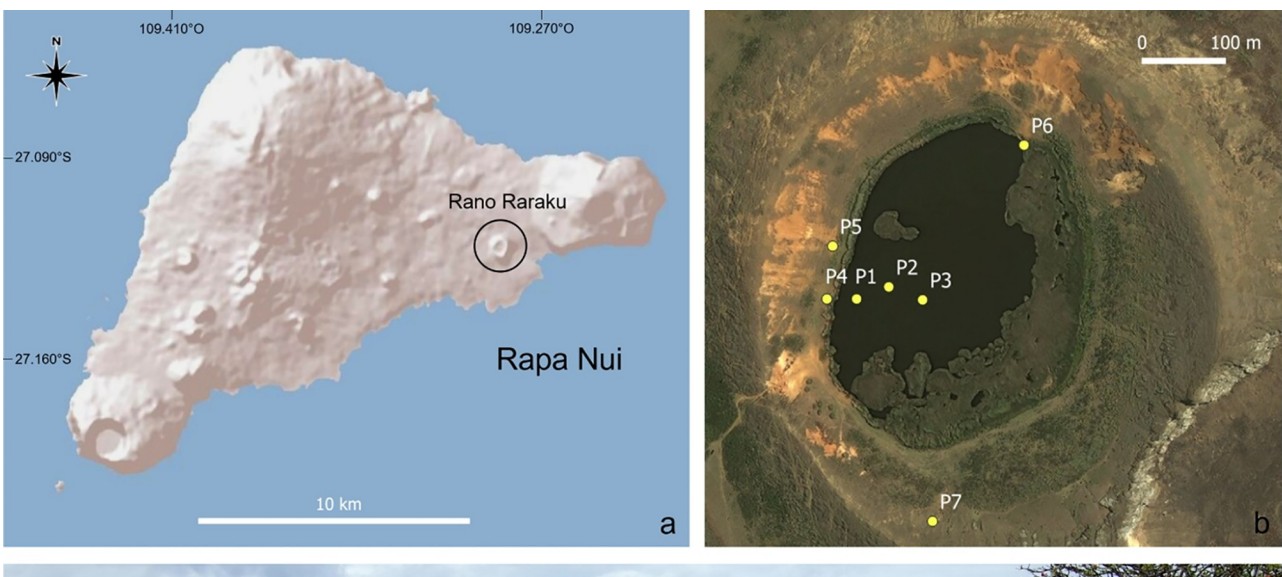

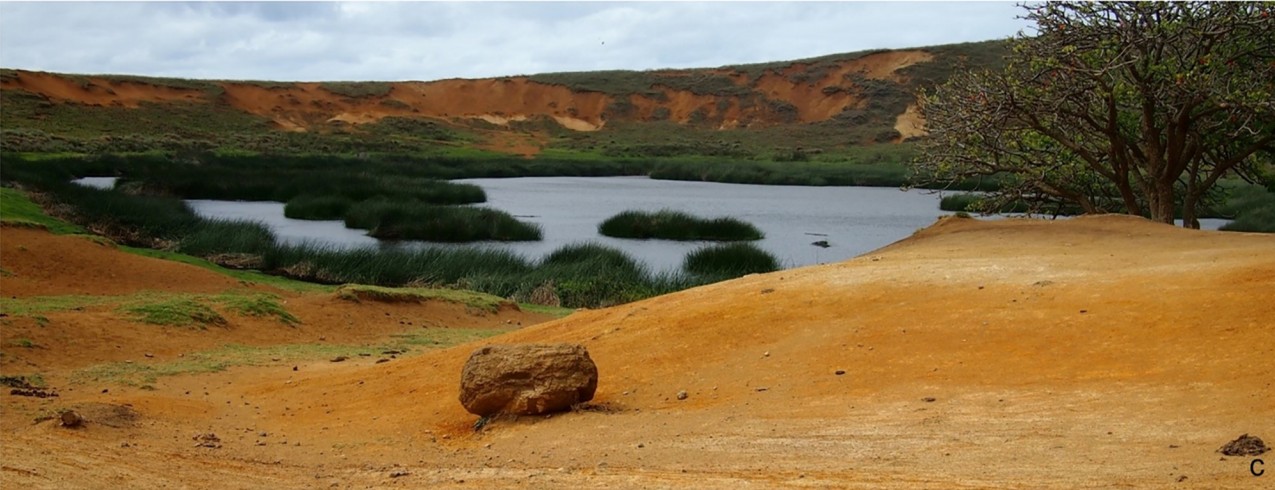

**Fig 1.** (a) The location of Rano Raraku on Rapa Nui (Easter Island). Base map and data from ESRI and OpenStreetMap Foundation. (b) Aerial view of the crater lake with the positions of the samples collected. Map provided by Base map and data from ESRI and OpenStreetMap Foundation. (c) Picture of the Rano Raraku basin on 7 Sep 2017. The photo was taken on the SW section of the lake and photo is taken by Dario Battistel facing NE.

of changes in land use and the introduction of semi-wild sheep, cattle, and horses around 1864 CE, where their grazing and associated erosion impacted the hydrological system.

## 2.2 Sample collection

Samples were collected in September 2017. All necessary permits for field activity in the Rano Raraku area were obtained from the Chilean Government through the Corporaciòn Nacional Forestal and the Consejo de Monumentos Nacionales de Chile (see the Acknowledgments section for details).

Three surface water samples (~500 mL) were collected at P1, P2, and P3 in September 2017 (Fig 1). At each of the three locations, P1-P3, water samples (~500 mL) were also collected from the water column at depths of 90 (P1), 51 (P2), and 38 (P3) cm, respectively. Six 4-cm thick topsoil samples (~5 g) were collected at P4, P5, P6, and P7 (Fig 1). Exact locations of the sampling points are reported in Fig 1, and Table 1 records the characteristics of each sample. A

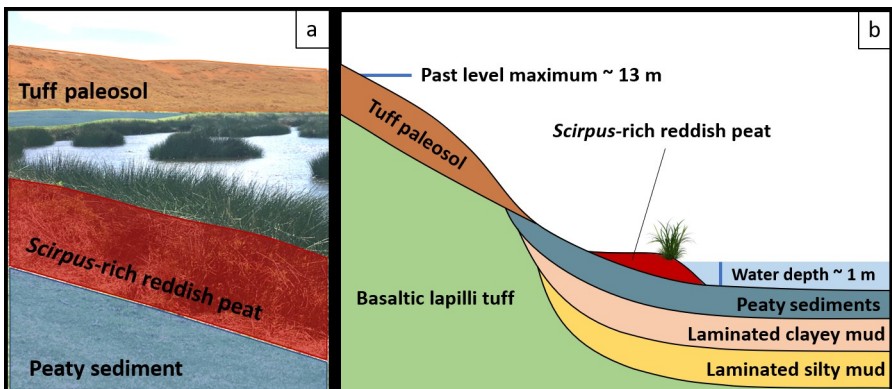

**Fig 2.** Geological representation of Rano Raraku a) picture of the lake (7 Sep 2017) with facies, b) cross section of the crater lake with the past lake level maximum at 17,000 cal yr. BP and at the time of the sampling [6–8, 10–12].

lake sediment core (LTS-RKU17-1A-1P-1), approximately 1 m long, was collected 95 cm below the water surface using a Livingstone-type surface corer [46] at P3. In this paper, we examine sediment samples corresponding to (a) the water-sediment interface (WSI; top 2 cm) and (b) bed sediments (between 3–5 cm depth).

## 2.3 Sample preparation and analysis

All water samples were filtered *in situ* through 0.45 μm cellulose nitrate membranes (Whatman) using a polycarbonate system equipped with a manual vacuum pump. Membranes containing suspended particulate matter (SPM) were stored in plastic Petri dishes, while the filtered water was kept in 50 mL polyethylene bottles. All samples were stored at 4˚C before being shipped to the Ca' Foscari University of Venice for analysis. Soil and sediment samples were dried to a constant weight in a desiccator at room temperature before further preparation for the analysis.

## 2.4 Water analysis

Major ions ($Cl^-$, $SO_4^{2-}$, $NO_3^-$, $NH_4^+$, $Na^+$, $K^+$, $Ca^{2+}$, and $Mg^{2+}$) in water samples were analyzed using an ion chromatography system (883 Basic IC plus, Metrohm) equipped with a Metrosep

**Table 1. Characteristics of samples collected at Rano Raraku.**

| Location | Samples | Depth (cm) | Elevation (masl) | Volume / Weight |
|---|---|---|---|---|
| P1 | Water–Suspended particulate matter (SPM) | 0 | - | 500 ml |
| | Deep water–SPM | 90 | - | 500 ml |
| | Water-sediment interface (WSI) | 0–2 | - | 5 g |
| | Bed sediment | 3–5 | - | 5 g |
| P2 | Water–SPM | 0 | - | 500 ml |
| | Deep water–SPM | 50 | - | 500 ml |
| P3 | Water–SPM | 0 | - | 500 ml |
| | Deep water–SPM | 38 | - | 500 ml |
| P4 | Topsoil | - | 82 | 5 g |
| P5 | Topsoil | - | 88.5 | 5 g |
| | Topsoil | | 89 | 5 g |
| P6 | Topsoil | - | 80 | 5 g |
| | Topsoil | | 81 | 5 g |
| P7 | Topsoil | - | 105 | 5 g |

Cation 1–2 (particle size 7 μm; eluent: $HNO_3$ 3 mM) and a Metrosep Anion supp/4 (particle size 5 μm; eluent: $HCO_3^-/CO_3^{2-}$ buffer 1.7/1.8 mM) column for cation and anion analysis, respectively. Liquid samples were directly injected into the chromatographic system after filtration through 0.45 μm syringe filters. Quantification of trace elements (TEs) and REEs in water samples were performed at the ISP-CNR laboratory in Venice (Italy) using an inductively coupled plasma—sector field mass spectrometer (ICP-SFMS model Element-XR, Thermo Scientific, Bremen, Germany). Water samples were acidified with ultrapure grade $HNO_3$ (Romil) and analyzed 24 hours later. TEs and REEs were quantified using external calibration curves. Method accuracy was assessed through the analysis of a certified reference material (TMRAIN95).

The isotopic composition of water samples was determined with a DeltaV-Advantage mass spectrometer (Thermo Scientific) equipped with a gas bench. For the analysis, 200 μL of each water sample was introduced into a glass vial. The samples were first flushed with a gas mixture of 2% $H_2$ in He and analyzed to determine $\delta^2H$. The same samples were then flushed with a gas mixture of 0.4% $CO_2$ in He and analyzed to determine $\delta^{18}O$ after 20 hours of equilibration. All samples were measured at least in triplicate. The SMOW2 and SLAP2 isotopic standards were used as references together with a laboratory standard that was analysed every nine samples to evaluate the stability of measurements. The isotopic compositions are expressed as $\delta^2H$ for $^2H/\,^1H$ or $\delta^{18}O$ for $^{18}O/^{16}O$ in the samples (s) and in the reference material (r), respectively.

## 2.5 Sediment, soil, and SPM analysis

Sediment, soil and SPM samples were digested with a mixture of $HNO_3$, HCl and HF (6:2:1 mL for 0.2 g of sample, suprapure grade acids, Romil), using an Ethos 1 microwave oven (Milestone) and a temperature-controlled program up to 200˚C in pressurized Teflon vessels. The digests were diluted in 50 mL of ultrapure water (PURELAB Ultra system, Elga, High Wycombe, UK), randomized and analyzed within 24 hours for determination of major, trace, and rare earth elements. The analysis was carried out by ICP-MS (iCAP RQ, Thermo Scientific) equipped with an ASX-560 autosampler (Teledyne Cetac Technologies), PFE cyclonic spray chamber at temperatures of 2.7˚C, sapphire injector, quartz torch, Ni cones, and 1550 W of plasma RF power. Triplicate acquisitions were performed in kinetic energy discrimination (KED)–high matrix mode (collision gas He). Quantification was obtained by external calibration with standards prepared by mixing the multi-elemental solutions IMS-101, IMS-102, and IMS-104 (UltraScientific) and using on-line spiked Rh as the internal standard. Accuracy was assessed by contextual mineralization and analysis of the certified reference materials NIMT/ UOE/FM/001 (peat) and BCR-667 (estuarine sediment).

Total organic carbon in sediments (TOC %) was determined with a Flash 2000 HT Elemental Analyzer (Thermo Scientific). A small amount of dry sample (~0.2 mg) was weighed and sealed into tin capsules, then introduced into the CHNO quartz reactor, composed of chromium oxide, copper, silvered cobalt, and cobalt oxide. The reactor temperature was set at 950˚C, while the column was held at 65˚C. Samples were analyzed in triplicate.

## 2.6 Statistics

All statistical analyses were performed using the RStudio software. Pearson correlation coefficients and Student's t-tests were run to analyse data correlation and similarity, respectively. All of the data discussed throughout the paper are available in an open access repository: https://data.mendeley.com/datasets/k93rp3p4pd/1).

## 3. Results and discussion

### 3.1 Source of the Rano Raraku water

The absence of inflow streams into the crater lake of Rano Raraku strongly implies that the input of new freshwater mainly derives from meteoric water. Rapa Nui is a small, isolated island in the middle of the Pacific Ocean where the nearest land mass is >2000 km away. Because Rapa Nui is close to the South Pacific High and outside the range of the Intertropical Convergence Zone, evaporation and convective activity from the surrounding ocean provide the main source of precipitation. Precipitation that directly derives from a marine source often contains higher concentrations of salts than does precipitation from evaporation and convection from terrestrial freshwater sources. Our results are consistent with this idea, strongly indicating precipitation originated from evaporation of ocean water. The major element concentrations in filtered water samples demonstrate that Cl and Na are particularly abundant in the dissolved fraction (mean concentrations of 968 ± 28 and 609 ± 12 mg L$^{-1}$, respectively). The Na/Cl mean ratio is 0.63, slightly higher—but comparable to—the reference Na/Cl ratio in seawater (0.56) [47]. Major ions (Cl$^-$, SO$_4^{2-}$, Na$^+$, K$^+$, Ca$^{2+}$, and Mg$^{2+}$) in Rano Raraku correlate (r$^2$ = 0.970; slope of 18 ± 1; *p*-value < 0.001) with typical seawater [47] providing further evidence of a direct marine source. The Na/Cl ratio in Rano Raraku water (0.63 ± 0.01) is also comparable to that reported in Hanga Roa meteoric waters collected in 2002–2003 (3–14 mg L$^{-1}$, 0.74 ± 0.12; *p*-value = 0.016; [42]). Therefore, Na$^+$ and Cl$^-$ input almost completely results from meteoric waters, in agreement with the general scarcity of Na$^+$ found in soils in the Rano Raraku catchment. This finding supports the results of Herrea and Custodio [42], who found that that the contribution from runoff and soil erosion is likely negligible.

The isotopic composition of Rano Raraku water ranges between 25 ± 2 ‰ and 4.0 ± 0.1 ‰ for δ$^2$H and δ$^{18}$O, respectively. These positive values indicate that the lake has predominantly been affected by evaporation processes. A simplified version of the Craig and Gordon [48] model that assumes no resistance to mixing in the liquid phase can model closed water systems (such as a small lake), where no inflow or outflow is present and the isotopic composition is driven only by evaporation [49]. The observed slope of the evaporation line (LEL) predicted by the Craig-Gordon model is:

$$S_{LEL} = \frac{\left[\frac{h(\delta_A - \delta_P) + (1+\delta_P)\left(\varepsilon_K + \frac{\varepsilon^+}{\alpha^+}\right)}{h - \varepsilon_K - \frac{\varepsilon^+}{\alpha^+}}\right]_{\delta D}}{\left[\frac{h(\delta_A - \delta_P) + (1+\delta_P)\left(\varepsilon_K + \frac{\varepsilon^+}{\alpha^+}\right)}{h - \varepsilon_K - \frac{\varepsilon^+}{\alpha^+}}\right]_{\delta^{18}O}} \tag{1}$$

where δ$_A$ is the isotopic composition of atmospheric moisture, δ$_P$ is the isotopic composition of precipitation, *h* is the relative humidity, α$^+$ is the liquid-vapor equilibrium isotopic fractionation, ε$^+$ is the equilibrium isotopic separation between liquid and vapor, and ε$_k$ is the equivalent kinetic isotopic separation based on tunnel experiments [42, 49–52]. The α$^+$ value is solely a function of temperature that, in this study, was assumed to be equal to 20˚C. This temperature is based on the fact that Rapa Nui temperatures do not change significantly during the year, because they only range between 18˚C in July/September and 23˚C in January/March [42]. Relative humidity was set to *h* = 0.79 [33]. Based on these assumptions, LEL was 6.02, slightly lower than the mean meteoric water line (δ$^2$H = 8·δ$^{18}$O + 10‰; blue line in Fig 3). Forcing the straight line through the δ$^2$H and δ$^{18}$O mean precipitation values, we obtained the equation δ$^2$H = 6.02·δ$^{18}$O + 4.04‰ (red line in Fig 3). The isotopic composition of the Rano Raraku water matches the model, as the points (in red) lie close to the red line.

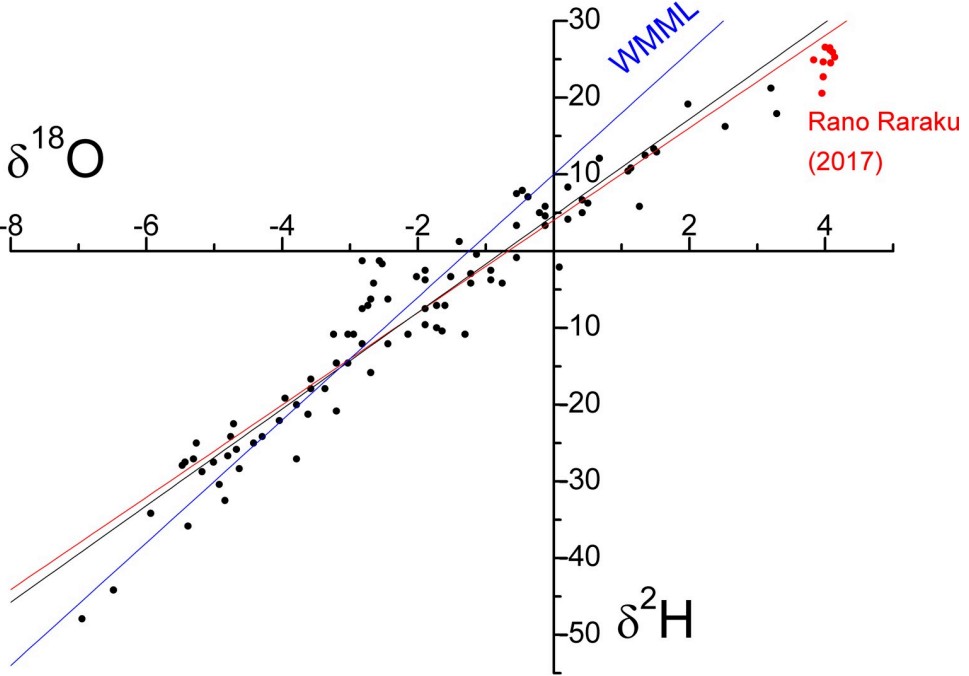

**Fig 3. Isotopic composition of rainfall on Rapa Nui (black dots; from Herrera and Custodio [42]) and Rano Raraku water in 2017 (red dots).** Mean meteoric water line (blue), rainfall best fit (black line), and equation straight line derived from the evaporation model (red line).

## 3.2 Chloride mass balance

To evaluate the precipitation-evaporation balance during the last few decades, we estimate the chloride mass balance. We assume that $Cl^-$ input predominantly derives from saline rainfall, while evaporation further increases the salinity of the lake water. In 2017, which was the driest year since 1967 (Center for Climate and Resilience Research, http://www.cr2.cl/datos-de-precipitacion/?cp_Precipitacion=1), the mean $Cl^-$ concentration in Rano Raraku lake water was 968 mg $L^{-1}$. Considering a mean lake depth of 0.6 m, a surface area of 0.07 km², and a mean slope of 10% (both calculated from aerial images of the basin at the time of the sampling), the total mass of chloride in the lake was ~38 metric tons. This number was obtained by approximating the lake to the frustum of a cone and multiplying the volume of liters by the measured concentration of chloride. As previously mentioned, the rainfall regime differs across Rapa Nui, where precipitation is more abundant in higher elevations including Mount Terevaka (507 masl) than in the rest of the island [42]. The only multi-decadal (1954–2018) precipitation record for Rapa Nui is from the Mataveri airport gauge in Hanga Roa (Fig 4). Although not representative of the overall precipitation regimes in the island, this site is comparable to Rano Raraku in terms of annual rainfall [53]. Based on the Mataveri rainfall data, mean annual precipitation at Rano Raraku can be estimated around 1100 mm $yr^{-1}$ with a $Cl^-$ content of ~7 mg $L^{-1}$ (derived from Herrera and Custodio [42]).

Under these assumptions, we estimate that the amount of $Cl^-$ input to the basin, provided solely by meteoric water, averaged 550 kg $y^{-1}$ (at 2017 lake level conditions), obtained by multiplying the annual volume of meteoric water deposited on the lake surface by the average chloride concentration in rainfall. The precipitation record was used to calculate the amount of $Cl^-$ provided solely by precipitation since 1954 and resulted in a total of approximately 35 metric tons, a value not far from the 38 metric tons of $Cl^-$ estimated in Rano Raraku in 2017.

In 1990, the electrical conductivity of the lake water was 640 μS cm$^{-1}$ (i.e. 0.64 dS m$^{-1}$[41]). Using the equation proposed by Peinado-Guevara et al. [54] ([Cl$^-$] (meq L$^{-1}$) = 4.928 EC (dS m$^{-1}$)), we can estimate a concentration of 112 mg L$^{-1}$. Considering a mean lake depth of ~3 m [41] and a surface area of 0.09 km$^2$ [3] in 1990, we can estimate that about 25 metric tons of Cl$^-$ were present in the lake at that moment. In this study, the precipitation record was also used to determine the amount of Cl$^-$ supplied by rainfall. Assuming the chloride content to be constant in rainfall (7 mg L$^{-1}$), the amount of the ion that accumulated in the lake since 1954 is ~25 metric tons, which coincides with the previous estimation supporting precipitation as the only source of chloride in the lake. The discrepancy between the 2017 estimates of chloride accumulated in the lake and the amount supplied by rainfall can likely be ascribed to the contribution of dry deposition. This source of chloride can be significant although it is rather difficult to model since it is influenced by several factors such as windspeed, elevation, and distance from the sea [55]. In Atlantic coastal sites, for example, dry deposition of chloride can reach up to 30% of the total bulk deposition [56]. The relative contribution of dry deposition is certainly increased with the pronounced decrease in precipitation that occurred from 2010 to 2017 (Fig 3) and, together with intensified evaporation, contributed to the high concentration of Cl$^-$ we measured in the Rano Raraku water.

These data from 1990 and 2017 suggest that Cl$^-$ only started to accumulate in the Rano Raraku lake since the 1950s, even though the lake formed thousands of years ago (>50,000 cal yrs BP). This accumulation could be due to the fact that Rano Raraku is a naturally closed basin and that the insertion of a pipe in 1958 [13] could have artificially drained much of the water that was present in the 1950-60s.

Although this calculation of chloride mass balance requires several assumptions, these results suggest that (a) the source of water in Rano Raraku is solely meteoric and (b) Rano Raraku is a closed system where the lake level is controlled mainly by the evaporation-precipitation balance. The isotopic composition of the water in 2017 agrees with the hydrological closed system model where the isotopic composition is affected solely by evaporation. During the last 30 years, the lake level was reported to remain between the depths of 2 and 4 m [3, 6, 8, 11, 41]; although current lake levels are less than 2 m. These fluctuations in water depth follow the same general trend as the precipitation record (Fig 4), further suggesting that the evaporation-precipitation balance is the main natural lake level control.

### 3.3 Redox conditions

In September 2017, Rano Raraku water levels were low (~1 m) and NH$_4^+$ concentrations were higher than nitrate concentrations in water, indicating a potential lack of nitrification reactions indicate highly anoxic conditions. In oxygenated waters, NH$_4^+$ is subject to nitrification, transforming NH$_4^+$ into N-oxidized forms, such as NO$_3^-$ [57]. At relatively low concentrations of dissolved oxygen, nitrification reactions of NH$_4^+$ ceases and consequently ammonium concentrations increase [58]. Besides the degradation of autochthonous organic matter, additional sources of NH$_4^+$ include fertilizers and animal feces. In recent times, the fecal input from free-roaming horses that often graze inside the crater may be a possible NH$_4^+$ source. Horses, as well as sheep and cattle, were introduced by the missionaries who settled in 1864 [13]. Although sheep were numerous for almost a century after 1870, they were then largely replaced by cattle and horses.

The higher NH$_4^+$ concentrations likely contribute to the slightly alkaline pH of 7.8 measured during our sampling campaign, which differs from the slightly acidic conditions (pH of 6.3) observed in 1990 by Geller [41]. At Rano Raraku, the total organic carbon content of the sediment samples was 18.3% and 16.9% in WSI and bed sediments, respectively. Organic

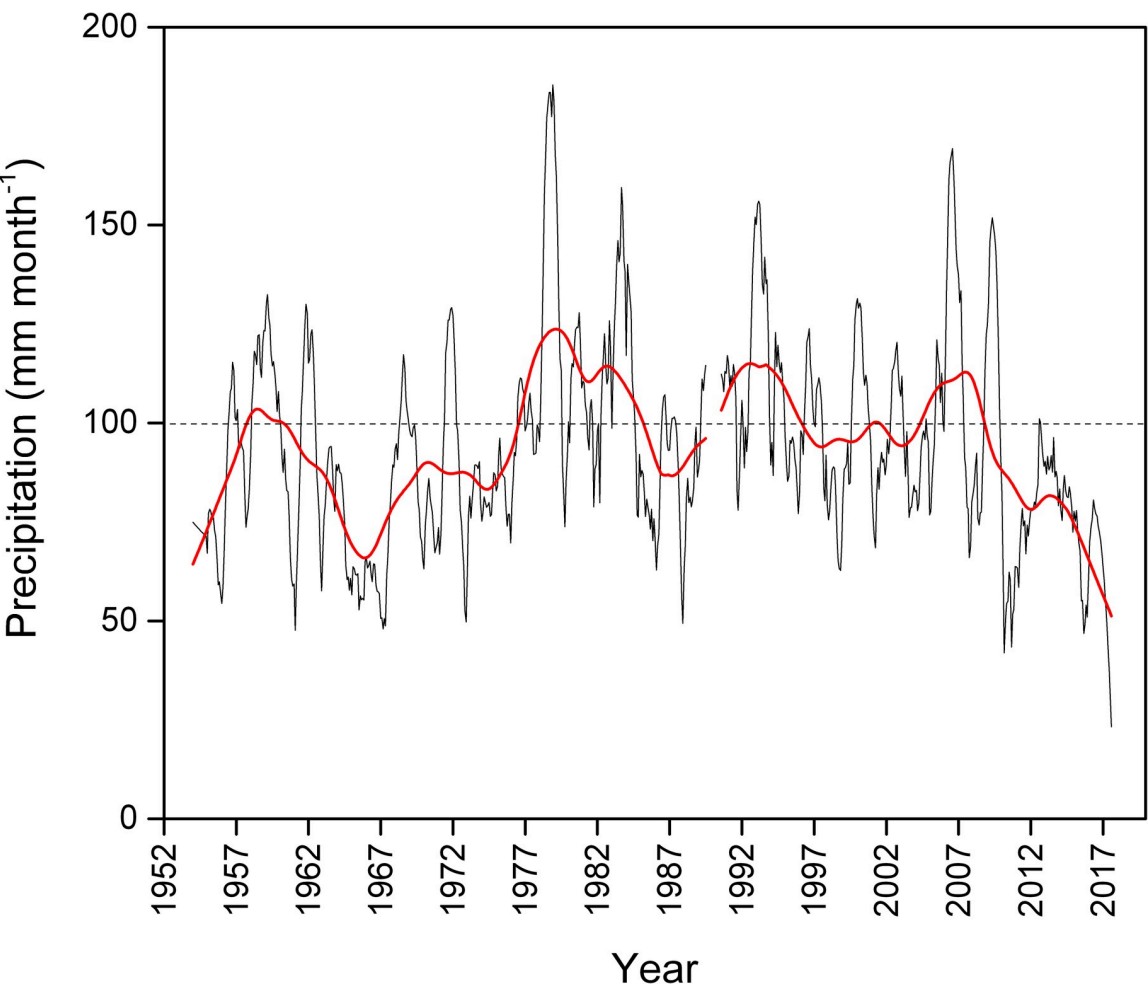

**Fig 4. Monthly precipitation at Mataveri, Rapa Nui (Easter Island) (Center for Climate and Resilience Research, http://www.cr2.cl/datos-de-precipitacion/?cp_Precipitacion=1).** The red line is a 10-year moving average.

decomposition can occur in both aerobic and anaerobic conditions, with a complex mechanism that involves several steps, favored by the presence of electron acceptors such as nitrates, sulfates, and $Fe^{3+}$. In water samples, sulphates were slightly depleted (i.e. $SO_4^{2-}/Cl^-$ ratio ~ $10^{-3}$) compared to the other shallow crater lakes fed by precipitation, where ratios are at least one order of magnitude higher [59]. This difference suggests that sulphate reduction in Rano Raraku might be involved in the decomposition of the organic matter.

In the aqueous phase, mean Mn and Fe concentrations were $4 \pm 1$ and $2.0 \pm 0.5$ µg $L^{-1}$, respectively, resulting in a relative enrichment of Mn (Mn/Fe = $2.0 \pm 0.9$). In the SPM, the concentrations of the redox-sensitive elements Mn and Fe were $58 \pm 18$ and $868 \pm 515$ µg $L^{-1}$, respectively. This large concentration range is related to the variability in SPM mass (data were normalized by the volume of water). The Mn/Fe was $0.07 \pm 0.02$ (with a variability less than 30%). The Fe and Mn concentrations were $96 \pm 14$ mg $g^{-1}$ and $2.3 \pm 0.4$ mg $g^{-1}$ in WSI (Fig 5). In bed sediments, Fe slightly increases while Mn substantially decreases. Consequently, the Mn/Fe ratio is higher in WSI ($0.024 \pm 0.003$) than in bed sediments ($0.015 \pm 0.002$). In the topsoil samples (Fig 5), Fe is particularly abundant in the form of oxyhydroxide minerals, as also observed by Horrocks et al. [11]. However, the topsoil Mn/Fe ratio is $0.020 \pm 0.005$ and is close to the Mn/Fe in the upper crust (i.e. 0.020; [60]) as well as the ratio in volcanic glass in Rano

Raraku hyalotuff (i.e. Mn/Fe ~ 0.025 ± 0.007; [45]), but substantially lower than values found in Rano Raraku water and in SPM.

The cycling of Mn and Fe in lake sediments is associated with the oxic/anoxic conditions of the water body [61] and is often used as a proxy in paleoclimate reconstructions [32, 62, 63]. External sources of Mn and Fe include catchment erosion and redox-related dissolution of minerals. Internal biochemical and/or physical processes are responsible for the increasing/ decreasing concentrations of these elements in the water column, as a result of remobilization and redox-controlled release from the sediment [64–66]. In Rano Raraku, the external input of Mn and Fe derives from catchment erosion or wind transport from the surrounding areas, where the Mn/Fe ratio in topsoil is 0.020 ± 0.005 on average, ranging from 0.017 to 0.029. Fig 6 presents a schematic explanation of the cycling of Mn and Fe at Rano Raraku. The reducing (anoxic) environment favors the reduction of Mn(IV) and Fe(III) in the sediment where Mn (II) and Fe(II) diffuse through the water body. Due to the higher mobility of Mn(II) ions, they diffuse faster than Fe(II), enriching the WSI with Mn (Mn/Fe = 0.024 ± 0.003), compared to the underlying sediment (Mn/Fe = 0.015 ± 0.002). In the water, Mn(II) is stabilized by the reducing conditions and prevails in dissolved forms (Mn/Fe = 2.0 ± 0.9). The reoxidation of Fe is faster than Mn, promoting the formation of iron oxyhydroxides that adsorb onto the SPM, where the SPM is then enriched in Fe compared to the dissolved fraction (Mn/ Fe = 0.07 ± 0.02), but more depleted than the sediment. This mechanism implies that under anoxic conditions, Mn/Fe is expected to increase in the water body, while decreasing in the sediment, as supported by concentration data (Fig 6).

In addition to the Mn/Fe ratio, Cr and V can also reconstruct redox conditions in lacustrine environments. Under anoxic and more reducing conditions, vanadyl ($VO^{2+}$) and Cr(III) ions interact differently with surface particles and organic ligands. $VO^{2+}$ prevails in dissolved forms, thus enriching the Cr/V ratio in sediments. In the aqueous phase, Cr and V concentrations were 61 ± 14 and 116 ± 39 ng $L^{-1}$, respectively, providing Cr/V values of 0.6 ± 0.2. In the SPM, Cr and V concentrations were 6.4 ± 0.7 and 2.2 ± 0.4 µg $L^{-1}$, respectively, resulting in a Cr/V ratio of 3.0 ± 0.2. The Cr/V ratio for SPM was significantly higher (p-value $< 10^{-4}$) than the dissolved fraction ratio. Cr/V ratios in WSI and bed sediments do not show substantial differences, although these values (Cr/V = 0.039 ± 0.006 and 0.038 ± 0.006) are considerably lower than Cr/V in all of the other environmental compartments, indicating a relative depletion of Cr with respect to V. Vanadium content in topsoil samples was 155 ± 62 µg $g^{-1}$ and, despite the high variability, does not significantly differ on average from the upper crustal concentration (p-value = 0.07; V = 97 µg $g^{-1}$ from Rudnick and Gao [60]). Conversely, the Cr concentration in soils is 64 ± 18 µg $g^{-1}$, which is significantly depleted when compared to the upper crust composition (Cr = 92 µg $g^{-1}$ from Rudnick and Gao [60]; p-value = 0.012). Therefore, the Cr/V value of 0.4 ± 0.1 in the soil is significantly lower than SPM but comparable to values found in water (p-value = 0.16).

Although Cr/V has lower values in the water (0.6 ± 0.2) and higher values in SPM (3.0 ± 0.2), the Cr/V ratio is very low in the WSI and in the underlying sediment. Since V is expected to form more soluble compounds in reducing environments, this occurrence in the sediment is unexpected. Vanadium equilibrium is sensitive to pH, and the speciation of V under reducing conditions is still under debate [67]. In reducing sediments, a mixture of vanadyl (IV) and unknown vanadium (III) species is strongly complexed by organic compounds, while under oxic conditions, vanadate (V) preferentially interacts with Fe (hydro)oxides. The Rano Raraku lacustrine system is particularly enriched both in Fe and in organic matter, and the pH increases with the eutrophication of the lake. Therefore, V may be entrapped in sediment, which is compatible with Cr/V values of 0.039 ± 0.006 in WSI and 0.038 ± 0.006 in the sediment underneath, meaning that using the Cr/V ratio as a proxy for redox conditions in Rano Raraku must be carefully evaluated.

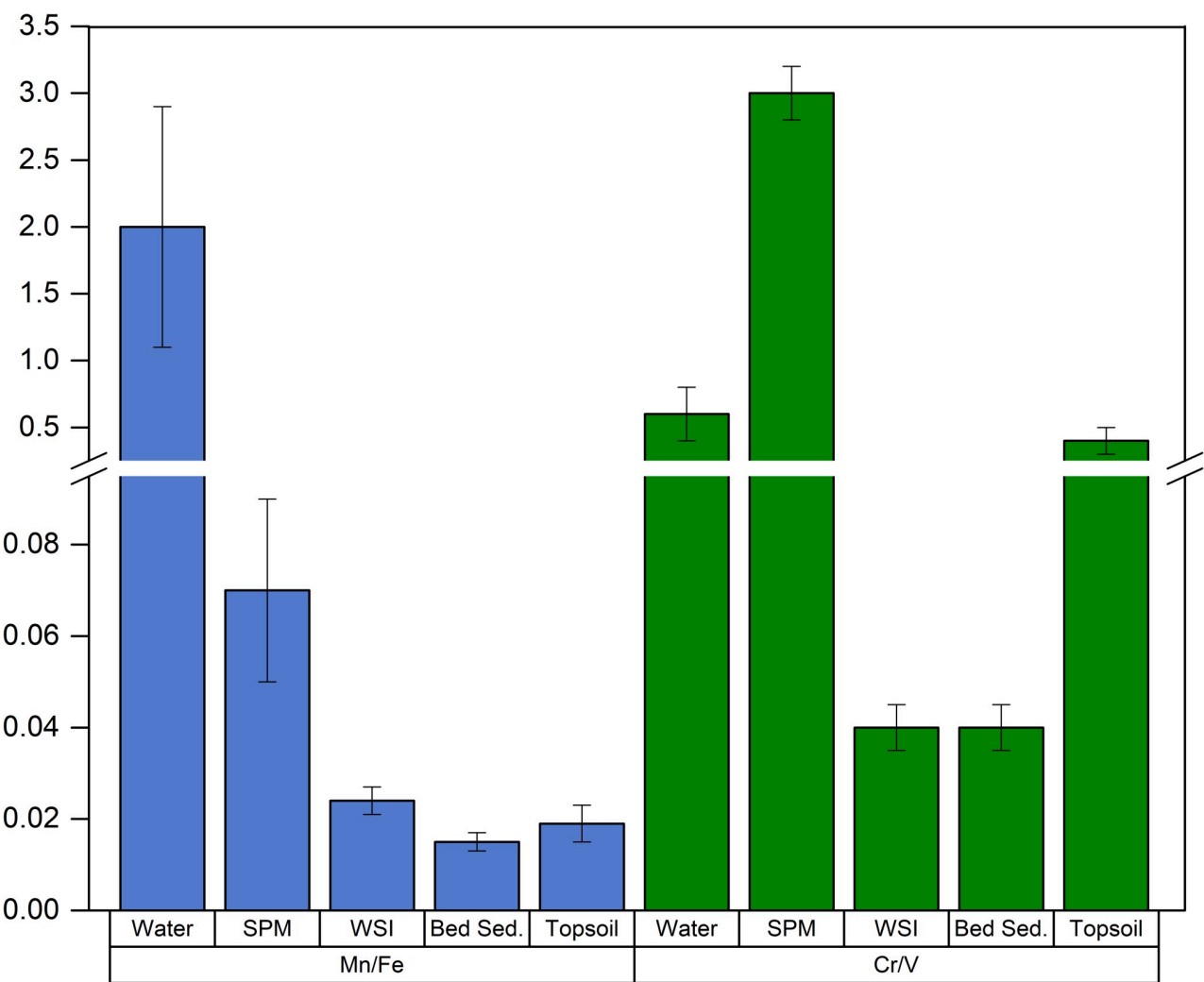

**Fig 5. Mean Mn/Fe (blue) and Cr/V (green) ratios with standard deviations of the analysed matrices: Water, SPM (suspended particulate matter), WSI (water-sediment interface), bed sed.** (bed sediment), topsoil.

### 3.4 Salinity and alkaline earth element fractionation

The Rano Raraku topsoil is generally depleted in alkaline earths and alkali metals, especially Na and Ca. The Ca/Sr and Mg/Ca ratios are 47 ± 15 and 8 ± 4, respectively. Although slightly higher than the ratios from SPM and water, the soil ratios are within the same order of magnitude.

The Rano Raraku water is relatively rich in Mg and Ca (mean values of 53 ± 1 and 15 ± 1 mg L$^{-1}$), followed by Sr (mean value 622 ± 136 µg L$^{-1}$), in line with other inactive crater lakes [59]. The Ca/Sr and Mg/Ca ratios are 24 and 3.6, respectively. Magnesium and Ca are strongly partitioned in the SPM (143 and 59 µg L$^{-1}$) compared to the more soluble alkaline ions (Na and K), which were lower than 20 µg L$^{-1}$. The SPM differs from the dissolved fraction and is depleted in Sr (2 ± 1 µg L$^{-1}$), but the Ca/Sr and Mg/Ca ratios (35 ± 6 and 2.5 ± 0.4) in SPM do not markedly differ from water. In the WSI, which is defined as the uppermost 2 cm of the core, alkaline earths are significantly higher (Ca 151 ± 23 µg g$^{-1}$; Mg 2.3 ± 0.3 mg g$^{-1}$) than in the deeper section (between 3 and 5 cm from the WSI, Ca 47 ± 7 mg g$^{-1}$; Mg 1.7 ± 0.3 mg g$^{-1}$), as well as Ca/Sr (WSI: 5.4 ± 0.8; bed sediments: 3.5 ± 0.5). However, Mg/Ca is higher in bed sediments (36 ± 5) than WSI (15 ± 2).

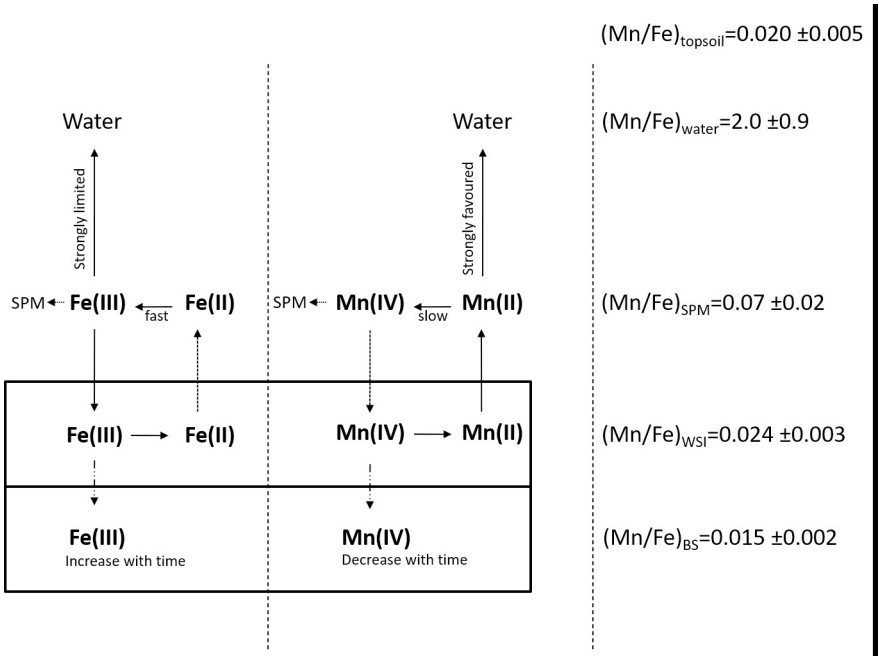

**Fig 6. Scheme of the fractionation process of Mn and Fe in Rano Raraku.**

The precipitation of aragonite and calcite infers oscillations in water salinity in marine and lacustrine systems [26, 28–30]. In more saline waters, high-Mg and high-Sr calcite or aragonite preferentially precipitate. The log-log plot of Mg/Ca *vs* Ca/Sr agrees with this process ($r^2$ = 0.983; *p*-value = 0.0089; Fig 7). Except in the topsoil, the increase in Mg/Ca from SPM to bed sediments corresponds with a decrease in Ca/Sr. Therefore, the precipitation of carbonates enriched in Mg and Sr depletes the water column of these elements, while the WSI is more depleted than in bed sediments, likely due to the mixing effect exerted by water. However, the topsoil is not directly involved in this fractionation process (Fig 7). As a first approximation, we assume that Mg/Ca (or Ca/Sr) responds linearly to a change in salinity, as observed in a previous study (Dissard et al., 2010).

## 3.5 REE fractionation and lake level oscillations

Average concentrations of REEs in Rano Raraku water range from 0.3 to 3.7 ng L$^{-1}$, where Ce and Nd are the most abundant. In this paper, we propose a diagnostic ratio between LREE (La, Pr, and Nd) and HREE (Er, Tm, Yb, and Lu) to investigate possible fractionation processes. The ratio LREE/HREE is defined as the following:

$$\left(\frac{LREE}{HREE}\right)_i = \frac{4 \cdot \left(\frac{[La]_i}{[La]_c} + \frac{[Pr]_i}{[Pr]_c} + \frac{[Nd]_i}{[Nd]_c}\right)}{3 \cdot \left(\frac{[Er]_i}{[Er]_c} + \frac{[Tm]_i}{[Tm]_c} + \frac{[Yb]_i}{[Yb]_c} + \frac{[Lu]_i}{[Lu]_c}\right)} \tag{2}$$

where the subscript *i* refers to the concentration in samples and the subscript *c* refers to the upper crust concentration [60]. As shown in Fig 8, the Rano Raraku water has LREE/HREE = 0.10 ± 0.02, indicating a depletion (enrichment) in LREE (HREE). Among the REEs, Ce is of particular interest as its concentration is affected by a combination of environmental factors and processes such as leaching, runoff, and redox equilibria. This combination is mainly due to the higher solubility of reduced Ce(III) compared to oxidized Ce(IV) species,

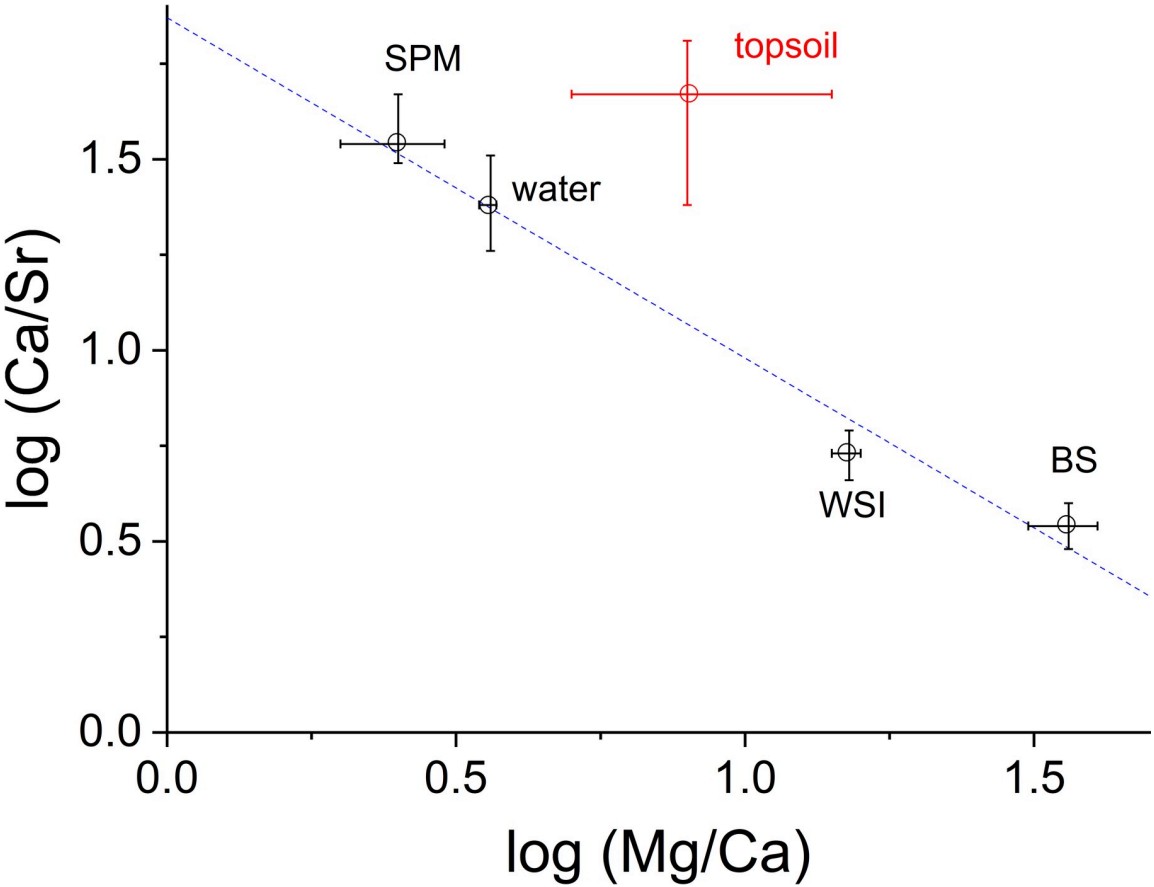

**Fig 7. Log-log plot of Mg/Ca vs Ca/Sr values in different environmental components.** Topsoil (red point) does not lie on the line because it is not involved in the fractionation process occurring in the lake.

which differs from the other REEs that exhibit only one redox state (III) in the environment (except Eu). The cerium anomaly (Ce/Ce*) is defined as the following [68]:

$$\frac{Ce}{Ce^*} = \frac{3 \cdot \left( \frac{[Ce]_i}{[Ce]_c} \right)}{2 \cdot \left( \frac{[La]_i}{[La]_c} \right) + \left( \frac{[Nd]_i}{[Nd]_c} \right)} \tag{3}$$

We also use Eq (3 to calculate Ce/Ce* in all of the investigated matrices. In water, Ce/Ce* was $0.6 \pm 0.1$, suggesting a depletion in Ce with reference to the value expected when no fractionation occurs (i.e. Ce/Ce* = 1).

In the Rano Raraku SPM, REE concentrations range between a few to hundreds of ng $L^{-1}$ where, similarly to the surrounding water, Ce and Nd are the most abundant REEs. The LREE/HREE ratio is $0.76 \pm 0.03$, which is significantly higher than in the dissolved phase ($p$-value $< 10^{-4}$), indicating a REE fractionation with an enrichment (depletion) of less (more) soluble LREEs (HREEs) in SPM. Similarly, Ce/Ce* $(1.02 \pm 0.04)$ is significantly higher in SPM than in water ($p$-value $< 10^{-4}$).

In sediments, REEs range between tens of ng to a few μg per gram, with higher concentrations of Ce, Nd, and La than the other REEs. The LREE/HREE ratio is $0.42 \pm 0.06$ and $0.28 \pm 0.04$ in WSI and bed sediments, respectively. The Ce anomaly slightly exceeds unity in

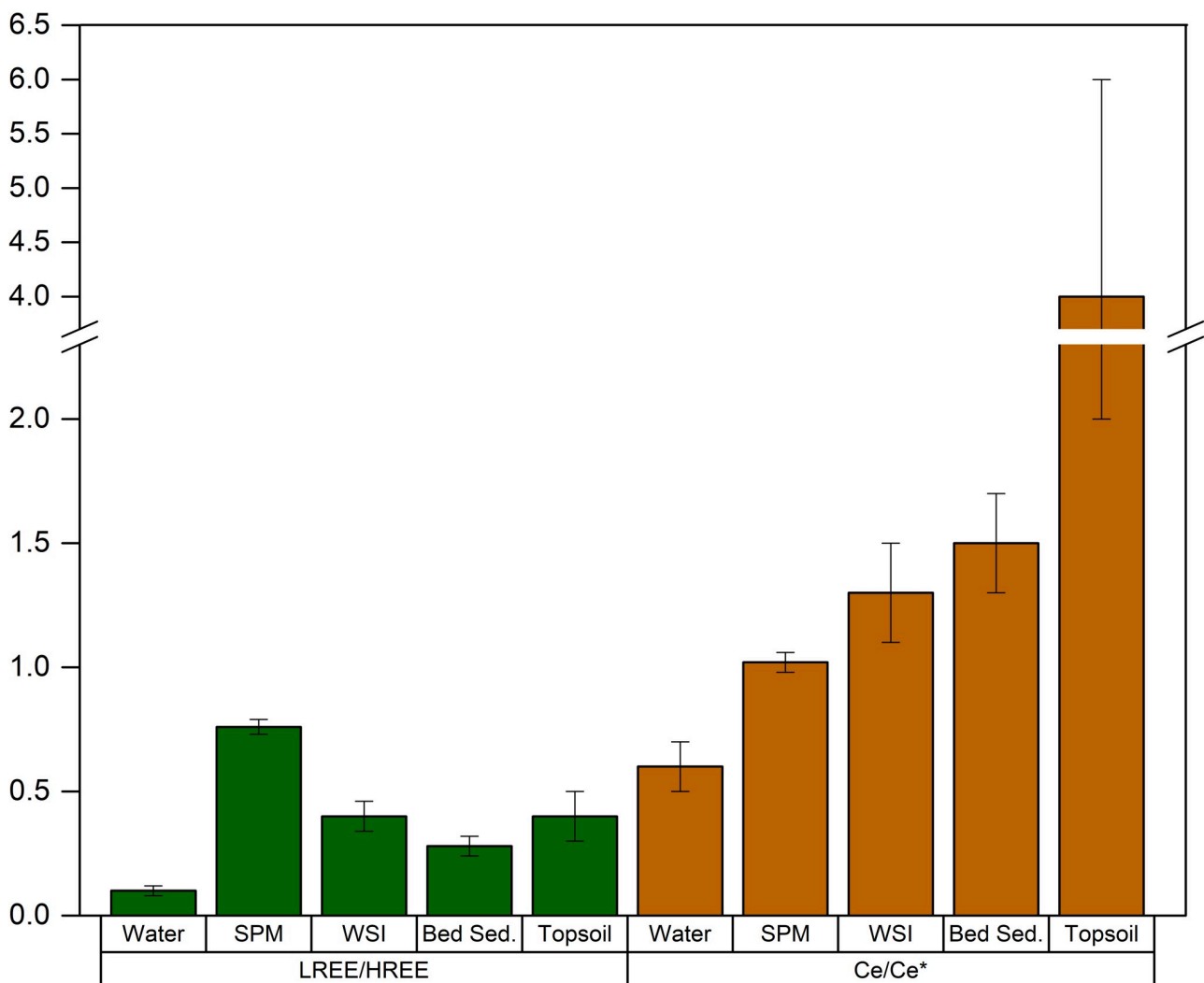

**Fig 8. Ce/Ce\* (brown) and LREE/HREE (green) mean ratios with standard deviation in the analyzed matrices: Water, SPM (suspended particulate matter), WSI (water-sediment interface), bed sed.** (bed sediment), topsoil.

both WSI (1.3 ± 0.2) and bed sediments (1.5 ± 0.2), indicating an enrichment in Ce, which is similar to the Ce anomaly in SPM.

Rare earth elements in topsoil samples contain concentrations between 1 and 150 µg g$^{-1}$ and are particularly enriched in Ce and to a lesser extent in Nd and La. The LREE/HREE ratio in soils was 0.4 ± 0.1, significantly higher than in the water, but lower than in SPM and comparable to ratios in the sediment. The cerium anomaly is more variable than in the other matrices with values (Ce/Ce\* = 4 ± 2) that are substantially higher than in both water and SPM, indicating a general excess of Ce in topsoil.

The REE fractionation is a complex process that can be due to leaching during runoff or the formation of soluble/insoluble compounds in the aquatic environment when organic or inorganic ligands are present. Assuming that the marine source contribution to REEs is negligible due to the low concentration of REEs in the Pacific Ocean [69], the comparison between LREE/HREE in the different compartments suggests that REEs fractionate inside the water body. This fractionation is driven by the difference in solubility between LREE and HREE in aqueous solutions,

where HREEs are more soluble than LREEs [70]. In Rano Raraku, a possible mechanism may involve the preferential LREE adsorption onto the particulate fraction bonded to Fe-Mn oxides or organic material or an enhanced solubility of HREEs due to the presence of organic anions, hydroxides, and carbonates. Although $Cl^-$ can increase LREE solubility [71], this mechanism does not seem to dominate in Rano Raraku despite the high $Cl^-$ concentration; in fact, LREE concentrations remain relatively low in water with respect to the other matrices.

Cerium can provide additional information on fractionation processes. In the water body, Ce/Ce* shows a slight depletion in the dissolved phase and a moderate enrichment in SPM and sediments. The reducing environment stabilized the relatively more soluble Ce(III) complexes, thus limiting the fractionation inside the water body. The slightly higher values in the underlying sediment, although not remarkable, may be ascribed to conditions in the recent past when the lake was slightly more oxic. Therefore, although Ce/Ce* fractionation resembles the LREE/HREE partitioning, the Ce anomaly may be potentially more sensitive to redox conditions than LREE/HREE, where the latter is more sensitive to the chemical composition of the water column. Therefore, REE analysis suggests that these elements are more sensitive to the dynamics occurring in the water body rather than to potential changes in the transport of exogenous material. Airborne transport has a potential role in terrigenous material input, as field observations confirm the presence of moderately intense dusty winds. Erosion or leaching processes during lake level fluctuations were likely limited during the last decades. However, all of these factors may have been more influential in the past, although their interpretation may be affected by contrasting factors. For example, if leaching increases during lake level rise, a Ce/Ce* depletion should be observed in the lake system, because Ce is less soluble than other REEs. Nevertheless, the lake level rise is accompanied by more oxic conditions that favor the Ce(IV) precipitation in the sediment. Therefore, the variation in Ce/Ce* in the sediment depends on the balance between these factors. In topsoil samples, LREE/HREE and Ce/Ce* significantly correlate when using a log-log scale ($r^2$ = -0.905; p-value = 0.013). LREE/HREE increases with increasing the elevation, while Ce/Ce* decreases (Fig 9). This trend is consistent with past lake level oscillations. At relatively higher lake levels, the topsoil may have been directly exposed to lake water only during times of exceptionally high lake levels, while at lower lake levels, the soil was in contact with water for a longer time. Lake water preferentially solubilizes HREE rather than LREE, and REE rather than Ce. Since these solubilization processes are kinetically controlled, the longer that soil was exposed to water, the lower the LREE/HREE and the higher the Ce/Ce*.

## 3.6 Managing freshwater limitations on Rapa Nui

The chloride mass balance demonstrated that the salinity of the lake is driven solely by evaporation-precipitation balance and chlorides are constantly introduced in the lacustrine system by saline rainfalls. The modern concentration of ~1 g $L^{-1}$ is a consequence of a combination of evaporation and precipitation that was reset when the lake was artificially drained during the 1950s. Sedimentary evidence shows how the basin has likely completely dried several times over the past four millennia. The last hiatus corresponds to approximately 4000 to 800 yr BP [11]. A high-end estimate of the chloride content of the lake could therefore be obtained by calculating the accumulation of chloride from the end of the hiatus (~1150 CE) to the crisis in population due to the introduction of epidemic diseases following European arrival (~1750 CE). With a lake level of 10 m, close to the maximum possible level estimated by Horrocks et al. [11], and in absence of further removal of water, the chloride concentration would be about 400 mg $L^{-1}$. This value is reasonably close to the 250 mg $L^{-1}$ threshold recommended today for potable water [72]. However, the concentration of 400 mg $L^{-1}$ likely exceeds the concentration at the time when people from Rapa Nui likely used Rano Raraku water for drinking,

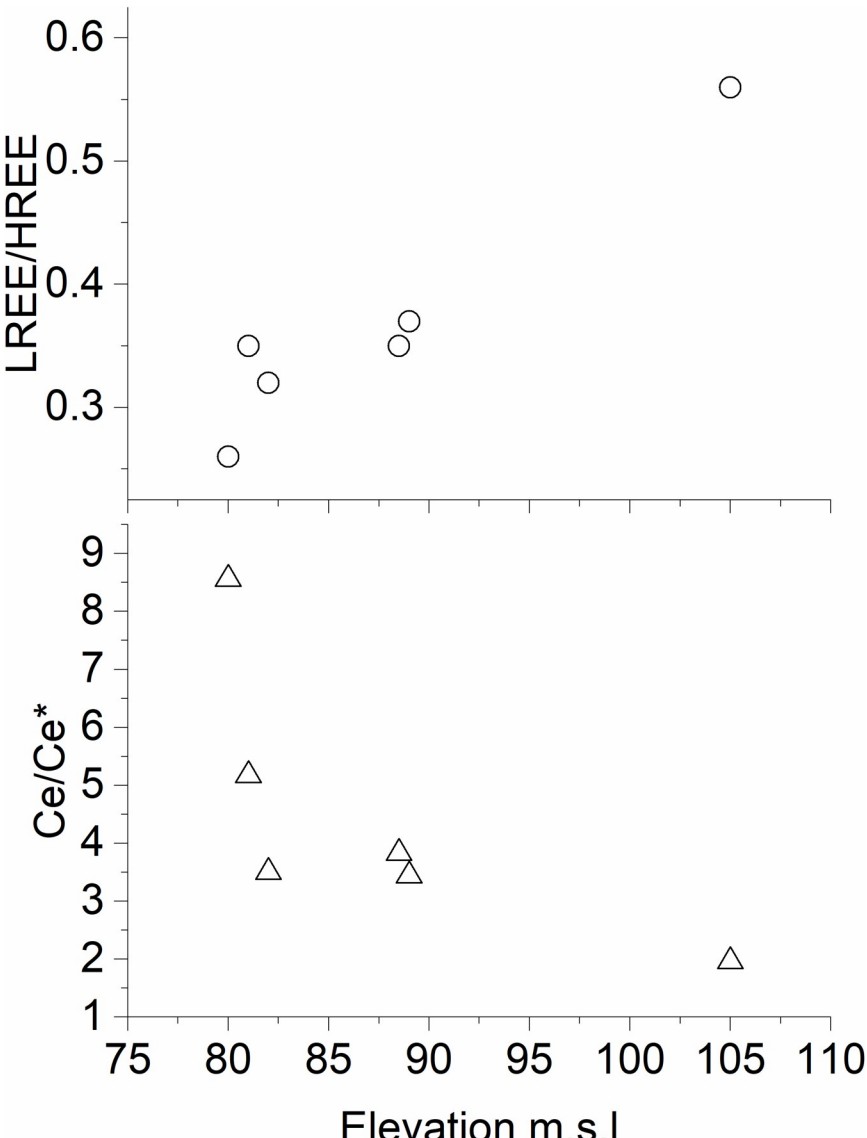

**Fig 9. LREE/HREE and Ce anomaly vs elevation of topsoil samples from the Rano Raraku internal crater.**

cooking, and irrigating including when a natural col was manually excavated and deepened by people [11]. Severe droughts are reported to have occurred between 1570 and 1720 CE [9, 17] causing the almost complete disappearance of the old forest surrounding Rano Raraku. This drought and associated vegetation change are believed to have promoted a significant cultural shift from the Ancient Moai to the Birdman cult from the mid-16[th] to late 18[th] centuries [73, 74]. The last catastrophic droughts on the pre-contact island coincided with the movement of the cultural centre to Orongo, near Rano Kau, where the Birdman cult existed until 1868, when slavery raids and foreign diseases brought Rapanui people to the brink of extinction.

The severe and prolonged droughts that are evident in the Rano Raraku record reflect a significant reduction of freshwater availability throughout the island. Undoubtedly, freshwater availability in Rapa Nui was, and still is, a considerable limitation. Archaeological evidence suggested that maintaining and managing limited freshwater resources on Rapa Nui was a

persistent challenge following human arrival. Historical accounts report that Rapanui used groundwater for drinking, in addition to the freshwater available in the lakes [75, 76]. DiNapoli et al. [77] recently found a significant correlation between the location of monumental platforms (*ahu*) and the presence of freshwater sources. Several groundwater seeps are present along the coastline 1–2 km from the Rano Raraku crater, radially discharging groundwater from the central aquifer [77]. We observed horses drinking freshwater along the coastline from these seeps near Rano Raraku. The Rapanui excavated large shallow wells along the lines of discharge of the central aquifer [78] as well as shallow artificial basins carved into basalt rocks (*taheta*) used to hold rainwater [79]. Many windmills were installed along the coast to pump water by the British company, Williamson-Balfour, which exploited the entire island of Rapa Nui as a sheep farm from 1895 to 1953 [80]. Until the late 1960s, Rapanui collected drinking water from deep reservoirs inside the many lava tubes near Roiho area, north of the main town of Hanga Roa. In the 1960s, deep wells were drilled to enhance modern aquifer exploitation for animals. One modern well is located close to Rano Raraku (about 0.7 km southwest from the cone and 0.9 km from the coast) and has a static water level of about 31 m, while the closest ancient well is located about 3.3 km from Rano Raraku along the northern coast and is 4 m deep. In 2010, the Chilean government published a hydrological survey of Rapa Nui. In this study, the $Na^+$ and $Cl^-$ content of the waters from both deep and shallow wells largely exceeded the Rano Raraku concentrations and increased with depth, further highlighting the issue of saltwater intrusion, well known by the Rapanui and worsened by modern exploitation [81]. Today, the salinity of groundwater in Rapa Nui ranges from ~80 mg $L^{-1}$ in the central Vaitea up to ~1.9 g $L^{-1}$ along the coast [42].

Norton [82] noted that Rapa Nui is the only major Polynesian island where drinking brackish water was a common occurrence. Thus, despite being salty, both the water from Rano Raraku and groundwater were likely an important source of drinking water for Rapanui. In addition to being brackish, the available water sources on Rapa Nui are limited and cannot sustain the water demand of the present population. In the past, even groundwater availability was significantly reduced all along the island, coinciding with prolonged droughts at Rano Raraku, potentially contributing the social instability in a densely populated island with an estimated population of up to 10–20 thousand people [53, 83–85].

## 4. Conclusions

Studies of current geochemical characteristics help interpret past environmental changes. Rapa Nui is often used as an example of a location where human influence on environmental change is closely linked and amplified due to Rapa Nui's size and isolation. However, few studies on Rapa Nui provide modern baselines against which these past changes can be compared. The interactions between soil, water, and lake sediments from Rano Raraku demonstrate a modern system that has substantially changed after the introduction of an outlet pipe in the 1950s and then the return to a closed basin after the 1970s. The stable isotopic ratios (25 ± 2 ‰ and 4.0 ± 0.1 ‰ for $\delta^2H$ and $\delta^{18}O$, respectively) and the Craig-Gordon model, demonstrate that the origin of Rano Raraku water is solely meteoric, originating from the nearby Pacific Ocean. This Pacific source is also highlighted by dissolved ion concentrations. The high salinity of Rano Raraku waters, due to the influence of coastal rainfall, affects the precipitation balance of alkaline earth elements. The topsoil is essentially depleted in alkaline earths, yet the uppermost 2 cm of the core contain relatively high alkaline earth concentrations (Ca 150 ± 20 µg $g^{-1}$; Mg 2.3 ± 0.3 mg $g^{-1}$) that are double those from the deeper section (between 3 and 5 cm from the WSI, Ca 50 ± 7 mg $g^{-1}$; Mg 1.7 ± 0.2 mg $g^{-1}$). The lake level is mainly dependent on the evaporation-precipitation balance, as Rano Raraku is essentially a closed system

with limited influence from runoff where the insertion of a pipe between 1950 until the end of the 1970s provided the only outflow. Rano Raraku is currently anoxic, as demonstrated by higher concentrations of ammonium than nitrate. Horses that graze on the shores of the lake may be responsible for the increasing ammonium concentrations since the 1990s, as well as the increase in lake pH since the 1990s.

Field observations and the balance of redox-sensitive elements such as Mn, Fe, Cr, and V are consistent with dominant reducing conditions in the basin, likely related to a current eutrophic state further supported by the high sedimentary organic carbon values. Ratios of Ce/Ce* are slightly depleted in the lake water yet are enriched in both SPM and sediments. The current reducing conditions stabilize the more soluble Ce(III) complexes. However, in the recent past, the lake was more oxic, resulting in slightly higher Ce/Ce* values in the underlying sediment. LREE/HREE values in the water and sediment agree with this change from past oxic to current reducing conditions, even though LREE/HREE is more sensitive to the chemical composition of the water column. The sensitivity of the lake to this human interference is important to consider when investigating past changes determined from Rano Raraku lake sediments as the Rano Raraku surrounding area was and has been an important cultural site and quarry for hundreds of years. Our results highlight how limited availability of freshwater resources on Rapa Nui was an enduring problem faced by Rapanui. Declines in the availability of freshwater resources may be linked to the decline of the Ancient Moai cult and is a problem that persists today.

## Acknowledgments

We thank the Corporación Nacional Forestal and, in particular, Lilian Gonzalez Nualart, Ninoska Cuadros, and Michel Pate. We also thank the Consejo de Monumentos Nacionales and, in particular, Maria Gabriela Atallah Leiva and Merahi Atam Lopez. Field and laboratory support was provided by CSDCO/LacCore, University of Minnesota. All the data collected for this research are available at Mendeley Data using the following link https://data.mendeley.com/datasets/k93rp3p4pd/1.

## Author Contributions

**Conceptualization:** E. Argiriadis, D. B. McWethy, D. Battistel.

**Data curation:** M. Roman, C. Turetta, D. B. McWethy, D. Battistel.

**Formal analysis:** M. Roman, C. Turetta, S. Hanif, D. Battistel.

**Funding acquisition:** N. M. Kehrwald, D. Battistel.

**Investigation:** N. M. Kehrwald, J. M. Ramirez Aliaga, D. B. McWethy, A. E. Myrbo, A. Pauchard, D. Battistel.

**Project administration:** A. Pauchard, D. Battistel.

**Resources:** J. M. Ramirez Aliaga, A. E. Myrbo, D. Battistel.

**Supervision:** E. Argiriadis, C. Barbante.

**Writing – original draft:** D. Battistel.

**Writing – review & editing:** E. Argiriadis, M. Bortolini, N. M. Kehrwald, M. Roman, C. Turetta, E. O. Erhenhi, J. M. Ramirez Aliaga, D. B. McWethy, A. E. Myrbo, A. Pauchard, C. Barbante, D. Battistel.

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
