## [Decision Letter · Decision Letter 0]

23 Mar 2021

PONE-D-20-39268

Easter Island Rano Raraku crater lake basin: geochemical characterization and implications for the Moai cult society

PLOS ONE

Dear Dr. Bortolini,

Thank you for submitting your manuscript to PLOS ONE. After careful consideration, we feel that it has merit but does not fully meet PLOS ONE’s publication criteria as it currently stands. Therefore, we invite you to submit a revised version of the manuscript that addresses the points raised during the review process.

Two reviewers have given a favorable opinion of the manuscript so I look forward to seeing a new version with minor revisions. My apologies for the delay in finding reviewers.

We look forward to receiving your revised manuscript.

Kind regards,

Greg Carling

Academic Editor

PLOS ONE

Journal Requirements:

1. 

3.  In your Methods section, please provide additional information regarding the permits you obtained to collect samples for the present study. Please ensure you have included the full name of the authority that approved the field site access and, if no permits were required, a brief statement explaining why.

Additional Editor Comments (if provided):

Reviewers' comments:

Reviewer's Responses to Questions

**Comments to the Author**

1. Is the manuscript technically sound, and do the data support the conclusions?

Reviewer #1: Yes

Reviewer #2: Yes

2. Has the statistical analysis been performed appropriately and rigorously? 

Reviewer #1: I Don't Know

Reviewer #2: Yes

3. Have the authors made all data underlying the findings in their manuscript fully available?

Reviewer #1: Yes

Reviewer #2: Yes

4. Is the manuscript presented in an intelligible fashion and written in standard English?

Reviewer #1: Yes

Reviewer #2: Yes

5. Review Comments to the Author

Reviewer #1: The manuscript refers mostly to climate related changes in the volcano crater lake. to My comments refer to the hydrological aspects and not to the paleoclimate results, which seems to me correct. Anthropic dominated situations like this are common in mountain wetlands.

The manuscript would benefit from an additional figure with a cross section of the volcano, the lake, the sediments (including the maximum level in the past), and the visible or inferred hydrogeological conditions in the flanks and in the surroundings. The Cl balance seem too coarse and the origin and value of Cl in rainfall is not mentioned. Under local conditions the values are quite variable and the dry atmospheric deposition should be added, as it is surely relevant. Lake water has the characteristics of rain and intra crater runoff evaporation, but rainfall data also show evaporation during the fall. The presentation of the water.chloride balance should be improved to make them more understandable (lines 265 and ....)

The age, alteration degree of the strato and pyroclastic volcano and erosion + organic sediments explain the existence of the lake, as in many other volcanic areas, but some comments should be added on the existence of a natural top drainage level and traces of possible springs or seeps in the outer cone surface and surroundings, if there are some observation ponts or excavations.

Line 56 Scoria-ceous delete -

Line 181 2H/1H add 1

Reviewer #2: The manuscript is well organized and well written making it easy to read. The authors presented a very good and detailed geochemical characterization of Rano Raraku freshwater lake. The authors used a very comprehensive and detailed experimental design to achieve their goal, which was to investigate the modern environmental process of the lake and its implications in past environmental changes and how those changes in the lake may have influenced in ancient cultures that inhabited on the island. The authors performed major ions, trace and rare earth elements, and diagnostic elements ratios analyses in different compartments of the lake: water (surface water and water column), sediment (sediment-water interface and bed sediments) and topsoil. They also performed an isotopic composition in the water. The materials and methods used are very complete and strong that supports the results found. I recommend the manuscript for publication.

This manuscript is an important contribution to the worldwide studies focused in freshwater systems, which in general are scarcer than marine ones. The only weakness of the work is that the authors begin the introduction directly by pointing to the lake under study. I think that before starting to talk about the lake under study, the authors should highlight the importance of the study carried out in Rano Raraku Lake in a more global context. What is known about this lake compared to other freshwater lakes in the world? (previous studies). Why is important to study the Rano Raraku lake? What is unknown about this lake that has not yet been studied previously?

The authors should highlight the importance of studying this lake compared to others. In lines 98-100 the authors name three previous studies in this lake “This study expands upon studies of major elements (Si, Ca, S, Fe and Ti) (Saez et al., 2009), pH and electrical conductivity (Geller, 1992) and macro and microfossils in a soil survey (Horrocks et al., 2012). The results of these studies led the authors to ask questions that led them to the present study?. What are those questions, if any? I wonder if the studies mentioned above served as a baseline for the development of this study. If so, I think that a briefly mention of the main results of those studies may contribute to understand better the importance of the present study.

Material and Methods

Well detailed and complete. I only suggest to the authors that they add what statistical analysis they used for the significant differences they show in the results and the statistical program used.

Line 146: “at -27.12220, -109.29031”: I suggest putting P1 in place of the coordinates to continue with the same writing format.

Line 158: “HDPE bottles”: What does HDPE means? Are special bottles?

Results and Discussion

Well developed and complete with reading easy to follow except in a few cases mentioned below:

Lines 219-220: “These values are within the range of the Eriksson (1952) equation that suggests Cl- values between 6 and 12 mg L-1 (in meteoric waters at ~1 km from the sea).” This sentence is a bit confusing for me. What does Ericksson's equation show? When the authors say these values are within the range… What values do the authors refer to? The values from table S4 correspond to the soil and the ones from Eriksson (1952) equation correspond to meteoric water. Also, Cl- values between 6 and 12 mg L-1 in meteoric waters are much smaller than found in the present study. Please clarify.

Line 247 “Chloride mass balance section”: This section is a bit confusing for me. In lines 271-274 the authors say “In this study the precipitation record was also used to determine the amount of Cl- supplied by rainfall. Assuming the chloride content to be constant in rainfall (8 mg L-1), the amount of the ion that accumulated in the lake since 1954 is ~ 29,000 kg”. Previously in lines 263-265 the authors said “The precipitation record was used to calculate the amount of Cl- provided solely by precipitation since 1954 and resulted in a total of approximately 40,000 kg”. I do not understand well what the difference between these two sentences is. It seems similar to me. In both cases the authors are using the precipitation record to calculate the amount of Cl supply by rainfall/precipitation since 1954. If that is not the case please clarify.

Also, in some cases is a bit difficult to follow how the authors made the calculations of Cl in this section. As examples:

Line 263 “we estimate that the amount of Cl- input to the basin, provided solely by meteoric water, averaged 680 kg y-1 (at 2017 lake level conditions)”. The value 680 kg y-1 comes from 41000kg/60years?

Line 265 “...of approximately 40,000 kg”: How is this value calculated?

Line 274 “...is ~ 29,000 kg”: the same here. Please clarify.

Lines 310-312 “In water samples sulfates were significantly depleted (i.e. SO42-/Cl- ratio ~ 10-3) compared to seawater (SO4 2-/Cl- = 0.14) suggesting that sulfate reduction may be involved in the decomposition of the organic matter”. Yes, but the authors should take into account that sulfate concentrations are typically two orders of magnitude higher in marine environments than in freshwater ones. I wonder if the difference in sulfate concentration between these two environments can affect the results. Is it possible to know if the difference found by the authors is not due to the lower concentration of sulphates in freshwater?

Line 324-325 “The Mn/Fe ratios are substantially lower than values found both in Rano Raraku water and in SPM (t-test of the means provides p-value lower than 10-4)”: Which is the meaning of this sentence? Please clarify. What Mn/Fe ratios do the authors refers to? Which samples? What statistical program was used for the analysis?

Line 361 “(p-value = 0.16)”: What statistical analysis and statistical program the authors used?

Line 398 “(r2 = 0.983; p-value = 0.0089”: What statistical analysis and statistical program the authors used?

Line 400-401 “Therefore, the precipitation of carbonates enriched in Mg and Sr depletes the water column of these elements”: I think this is contradicted by line 385 where the authors say that the water is enriched in Mg. Please clarify.

Line 424 “(Elderfield et al, 1982)”: Is this the correct form? Two authors are listed in the references. Please clarify.

Line 425 “∗”: What does it mean?

Line 432 “(p-value < 10-4)”: What statistical analysis and statistical program the authors used?

Line 435 “(p-value < 10-4)”: What statistical analysis and statistical program the authors used?

Line 476 “(r2 = -0.905; p-value = 0.013)”: What statistical analysis and statistical program the authors used?

Line 510 “Rull et al. 2016b”: Is this the correct form? Only one author is listed in the references. Please clarify.

Line 522 “Di Napoli et al. 2019”: Is this the correct form? 2018 is listed in the references. Please clarify.

Line 582 “References”: The authors should check the list references. See other minor comments in the text.

6. PLOS authors have the option to publish the peer review history of their article (what does this mean?). If published, this will include your full peer review and any attached files.

Reviewer #1: No

Reviewer #2: No

---

## [Author Response · Author response to Decision Letter 0]

7 Jun 2021

Point by point response to reviewers

Reviewer #1: The manuscript refers mostly to climate related changes in the volcano crater lake. to My comments refer to the hydrological aspects and not to the paleoclimate results, which seems to me correct. Anthropic dominated situations like this are common in mountain wetlands. 

The manuscript would benefit from an additional figure with a cross section of the volcano, the lake, the sediments (including the maximum level in the past), and the visible or inferred hydrogeological conditions in the flanks and in the surroundings. 

We added this figure in the text to better characterize the lake in the Materials and Methods | Study Site section. Figure 2 represents a cross section of the basin and a view of the internal Rano Raraku crater. 

The Cl balance seem too coarse and the origin and value of Cl in rainfall is not mentioned. Under local conditions the values are quite variable and the dry atmospheric deposition should be added, as it is surely relevant. Lake water has the characteristics of rain and intra crater runoff evaporation, but rainfall data also show evaporation during the fall. The presentation of the water chloride balance should be improved to make them more understandable (lines 265 and ....)

We refined the Cl- balance by adjusting some estimates to be compliant with source data and considering the slope of the lake shore to compute the water volume. Reference for the origin of Cl- concentration in rainfall was included. 

Dry deposition for chloride is seldom estimated in the literature, where mainly bulk deposition data is available. In addition, the rate of deposition is dependent on many factors such as wind, slope, elevation, distance from the sea. However, it is true that the input from dry deposition is relevant, accounting for up to the 30% in Atlantic coastal sites (Alcalà and Custodio, 2008). We included the contribution of dry deposition to account for the discrepancy between the total amount of chloride estimated in the basin and the value accumulated through rainfall. 

The age, alteration degree of the strato and pyroclastic volcano and erosion + organic sediments explain the existence of the lake, as in many other volcanic areas, but some comments should be added on the existence of a natural top drainage level and traces of possible springs or seeps in the outer cone surface and surroundings, if there are some observation ponts or excavations.

Several coastal seeps are present 1-2 km from the cone (DiNapoli et al., 2019), and both ancient and modern wells mainly fed by groundwater with a significant saline intrusion from the sea, increasing with depth, as reported by Herrera and Custodio (2008) and by the hydrological survey conducted for the Chilean government in 2010 (https://snia.mop.gob.cl/sad/SUB5214.pdf). The shallow wells were used by the ancient island inhabitants as water supply together with water from Rano Raraku and other crater lakes. We mentioned this in paragraph 3.6 “Anthropological implications”, which was reviewed to include this discussion and a more realistic estimation of chloride in Rano Raraku at the time of early settlers. 

Line 56 Scoria-ceous delete 

We corrected it with: “Scoria basaltic lapilli…”

Line 181 2H/1H add 1 

Reviewer #2: The manuscript is well organized and well written making it easy to read. The authors presented a very good and detailed geochemical characterization of Rano Raraku freshwater lake. The authors used a very comprehensive and detailed experimental design to achieve their goal, which was to investigate the modern environmental process of the lake and its implications in past environmental changes and how those changes in the lake may have influenced in ancient cultures that inhabited on the island. The authors performed major ions, trace and rare earth elements, and diagnostic elements ratios analyses in different compartments of the lake: water (surface water and water column), sediment (sediment-water interface and bed sediments) and topsoil. They also performed an isotopic composition in the water. The materials and methods used are very complete and strong that supports the results found. I recommend the manuscript for publication.

This manuscript is an important contribution to the worldwide studies focused in freshwater systems, which in general are scarcer than marine ones. The only weakness of the work is that the authors begin the introduction directly by pointing to the lake under study. I think that before starting to talk about the lake under study, the authors should highlight the importance of the study carried out in Rano Raraku Lake in a more global context. What is known about this lake compared to other freshwater lakes in the world? (previous studies). Why is important to study the Rano Raraku lake? What is unknown about this lake that has not yet been studied previously? 

The authors should highlight the importance of studying this lake compared to others. In lines 98-100 the authors name three previous studies in this lake “This study expands upon studies of major elements (Si, Ca, S, Fe and Ti) (Saez et al., 2009), pH and electrical conductivity (Geller, 1992) and macro and microfossils in a soil survey (Horrocks et al., 2012). The results of these studies led the authors to ask questions that led them to the present study?. What are those questions, if any? I wonder if the studies mentioned above served as a baseline for the development of this study. If so, I think that a briefly mention of the main results of those studies may contribute to understand better the importance of the present study.

A wide number of studies have been conducted at Rapa Nui, specifically at Rano Raraku, to investigate past environmental conditions in the past, to try to reconstruct the vegetation and climatic oscillation. Nevertheless, there is a lack of knowledge about the present and recent (∼ 800 yr BP) conditions of the lake.

We emphasized the importance of the results in the mentioned studies, highlighting their contribution to the knowledge of the past environmental and climatic conditions of Rapa Nui. The present work aims at investigating the modern environmental processes at Rano Raraku and put the basis for the application of trace element analysis to draw past environmental changes.

Material and Methods

Well detailed and complete. I only suggest to the authors that they add what statistical analysis they used for the significant differences they show in the results and the statistical program used. - 

The statistical analysis was performed using R software. We added a sentence in the Methods section: “All statistical analyses were performed using the RStudio software. Pearson correlation coefficients and Student’s t-tests were run to analyse data correlation and similarity, respectively.”

Line 146: “at -27.12220, -109.29031”: I suggest putting P1 in place of the coordinates to continue with the same writing format. 

We removed the coordinates and replaced that point with “P3” with the same coordinates.

Line 158: “HDPE bottles”: What does HDPE means? Are special bottles? 

We meant high density polyethylene bottles, for simplicity we used “polyethylene bottles” in the text.

Results and Discussion

Well developed and complete with reading easy to follow except in a few cases mentioned below:

Lines 219-220: “These values are within the range of the Eriksson (1952) equation that suggests Cl- values between 6 and 12 mg L-1 (in meteoric waters at ~1 km from the sea).” This sentence is a bit confusing for me. What does Ericksson's equation show? When the authors say these values are within the range… What values do the authors refer to? The values from table S4 correspond to the soil and the ones from Eriksson (1952) equation correspond to meteoric water. Also, Cl- values between 6 and 12 mg L-1 in meteoric waters are much smaller than found in the present study. Please clarify

We understand that this passage might have created confusion in the reader, so we preferred to remove the sentence and the reference to the Eriksson equation from the text. We completed the sentence about soil values by adding the following clarification: “[...] meaning that the contribution from runoff and soil erosion is likely negligible, as stated by Herrera and Custodio (2008).”

Line 247 “Chloride mass balance section”: This section is a bit confusing for me. In lines 271-274 the authors say “In this study the precipitation record was also used to determine the amount of Cl- supplied by rainfall. Assuming the chloride content to be constant in rainfall (8 mg L-1), the amount of the ion that accumulated in the lake since 1954 is ~ 29,000 kg”. Previously in lines 263-265 the authors said “The precipitation record was used to calculate the amount of Cl- provided solely by precipitation since 1954 and resulted in a total of approximately 40,000 kg”. I do not understand well what the difference between these two sentences is. It seems similar to me. In both cases the authors are using the precipitation record to calculate the amount of Cl supply by rainfall/precipitation since 1954. If that is not the case please clarify.

We used the precipitation record and average Cl concentration in rainfall twice, to calculate the accumulation of Cl in the lake from 1954 to 1990 and from 1954 to 2017, respectively, in order to show how the input is solely controlled by atmospheric deposition and the lake balance only depends on the precipitation-evaporation equilibrium.

Also, in some cases is a bit difficult to follow how the authors made the calculations of Cl in this section. As examples:

Line 263 “we estimate that the amount of Cl- input to the basin, provided solely by meteoric water, averaged 680 kg y-1 (at 2017 lake level conditions)”. The value 680 kg y-1 comes from 41000kg/60years?

Line 265 “...of approximately 40,000 kg”: How is this value calculated?

Line 274 “...is ~ 29,000 kg”: the same here. Please clarify.

We specified how values were obtained and added reference for the average chloride content in rainfall. Some values were corrected based on more precise estimates of the lake volume and Cl concentrations. 

Lines 310-312 “In water samples sulfates were significantly depleted (i.e. SO42-/Cl- ratio ~ 10-3) compared to seawater (SO4 2-/Cl- = 0.14) suggesting that sulfate reduction may be involved in the decomposition of the organic matter”. Yes, but the authors should take into account that sulfate concentrations are typically two orders of magnitude higher in marine environments than in freshwater ones. I wonder if the difference in sulfate concentration between these two environments can affect the results. Is it possible to know if the difference found by the authors is not due to the lower concentration of sulphates in freshwater?

We compared SO42-/Cl- ratios with waters from similar crater lakes fed by precipitation, rather than with seawater. Values in Rano Raraku are still at least one order of magnitude lower, thus we hypothesize a possible role of sulfates in the decomposition of organic matter. 

Line 324-325 “The Mn/Fe ratios are substantially lower than values found both in Rano Raraku water and in SPM (t-test of the means provides p-value lower than 10-4)”: Which is the meaning of this sentence? Please clarify. What Mn/Fe ratios do the authors refers to? Which samples? What statistical program was used for the analysis?

We connected the sentence to the former one, in order to make it clear that the Mn/Fe ratios referred to topsoil samples. We removed the reference to the t-test, since it was out of place. The statistical tools used were included in the methods section. 

Line 361 “(p-value = 0.16)”: What statistical analysis and statistical program the authors used?

Line 398 “(r2 = 0.983; p-value = 0.0089”: What statistical analysis and statistical program the authors used?

As mentioned above, the information about statistical analysis was included in the Methods section.

Line 400-401 “Therefore, the precipitation of carbonates enriched in Mg and Sr depletes the water column of these elements”: I think this is contradicted by line 385 where the authors say that the water is enriched in Mg. Please clarify.

Compared to other lake waters, Rano Raraku water is relatively rich in Mg and Sr, although in line with other crater lakes. We corrected the text and added reference. 

Line 424 “(Elderfield et al, 1982)”: Is this the correct form? Two authors are listed in the references. Please clarify. 

We corrected using “Elderfield and Greaves, 1982”.

Line 425 “∗”: What does it mean? – 

We use the expression Ce/Ce* to define the Cerium anomaly, as references in the paper use the same nomenclature. In the paper we used the equation from Elderfield and Greaves (1982) to point out the different Ce concentration into the analysed matrices.

Line 432 “(p-value < 10-4)”: What statistical analysis and statistical program the authors used?

Line 435 “(p-value < 10-4)”: What statistical analysis and statistical program the authors used?

Line 476 “(r2 = -0.905; p-value = 0.013)”: What statistical analysis and statistical program the authors used?

Please see above.

Line 510 “Rull et al. 2016b”: Is this the correct form? Only one author is listed in the references. Please clarify.

We corrected using “Rull 2016b”.

Line 522 “Di Napoli et al. 2019”: Is this the correct form? 2018 is listed in the references. Please clarify. – 

The correct reference is from 2019. We revised the reference section.

Line 582 “References”: The authors should check the list references. See other minor comments in the text.

References have been entirely checked in the text and in the reference list.

---

## [Decision Letter · Decision Letter 1]

6 Jul 2021

Rapa Nui (Easter Island) Rano Raraku crater lake basin: geochemical characterization and implications for the Moai cult society

PONE-D-20-39268R1

Dear Dr. Bortolini,

We’re pleased to inform you that your manuscript has been judged scientifically suitable for publication and will be formally accepted for publication once it meets all outstanding technical requirements.

Kind regards,

Greg Carling

Academic Editor

PLOS ONE

Additional Editor Comments (optional):

Reviewers' comments:

Reviewer's Responses to Questions

**Comments to the Author**

1. If the authors have adequately addressed your comments raised in a previous round of review and you feel that this manuscript is now acceptable for publication, you may indicate that here to bypass the “Comments to the Author” section, enter your conflict of interest statement in the “Confidential to Editor” section, and submit your "Accept" recommendation.

Reviewer #1: All comments have been addressed

Reviewer #2: (No Response)

2. Is the manuscript technically sound, and do the data support the conclusions?

Reviewer #1: Yes

Reviewer #2: (No Response)

3. Has the statistical analysis been performed appropriately and rigorously? 

Reviewer #1: Yes

Reviewer #2: (No Response)

4. Have the authors made all data underlying the findings in their manuscript fully available?

Reviewer #1: (No Response)

Reviewer #2: (No Response)

5. Is the manuscript presented in an intelligible fashion and written in standard English?

Reviewer #1: (No Response)

Reviewer #2: (No Response)

6. Review Comments to the Author

Reviewer #1: no new specific comments. I agree with the authors thar dry deposition is raraly measured as such but as a difference between bulk deposition and only wet deposition. My mention of dry depossitio was only to call the attention that bulk depossition may be significantly larger than only wet deposition as deduces from point samples of rainfall in gauges that only open prior to rain or are frequently washed,

Reviewer #2: (No Response)

7. PLOS authors have the option to publish the peer review history of their article (what does this mean?). If published, this will include your full peer review and any attached files.

Reviewer #1: No

Reviewer #2: No

---

## [Editor Report · Acceptance letter]

15 Sep 2021

PONE-D-20-39268R1 

Rapa Nui (Easter Island) Rano Raraku crater lake basin: geochemical characterization and implications for the Ahu-Moai Period 

Dear Dr. Bortolini:

I'm pleased to inform you that your manuscript has been deemed suitable for publication in PLOS ONE. Congratulations! Your manuscript is now with our production department. 

Kind regards, 

on behalf of

Dr. Gregory Carling 

Academic Editor

PLOS ONE